# Ideological Consistency and News Sharing as Predictors of Masking Among College Students

**DOI:** 10.3390/ijerph21121652

**Published:** 2024-12-11

**Authors:** Adrienne Holz

**Affiliations:** School of Communication, Virginia Tech, Blacksburg, VA 24061, USA; holz@vt.edu

**Keywords:** COVID-19, face masks, ideological consistency, compassion, self-esteem, fake news awareness, misinformation, partisan affiliation, political attitudes, media use

## Abstract

During the COVID-19 pandemic, the United States Centers for Disease Control and Prevention (CDC) recommended the use of well-fitting face masks or respirators as a strategy to reduce respiratory transmission; however, acceptance and utilization of face masks quickly became a contentious, politically charged matter. Given the effectiveness of masking against respiratory viruses, it is critical to understand the various normative factors and personal values associated with mask wearing. To this end, this study reports the findings of an online, cross-sectional survey (*n* = 1231) of college students during the COVID-19 pandemic. Findings show that ideological consistency, sharing news to create awareness, and sharing unverified news significantly predict masking behaviors, though ideological consistency most substantially explained variance in self-reported masking behaviors. Participants with more liberal political ideologies reported greater adherence to masking policies while those with more conservative ideologies reported less mask-wearing behavior. A better understanding of the predictors of masking behaviors, particularly how political ideologies continue to shape public health responses, is essential for designing more effective communication strategies to control disease spread and help inform strategies for future outbreaks. Study implications and limitations are discussed.

## 1. Introduction

The COVID-19 pandemic swept the world in early 2020. Caused by the severe acute respiratory syndrome coronavirus-2 (SARS-CoV-2), COVID-19 originated in Wuhan, China, in December 2019. The World Health Organization (WHO) characterized it as a pandemic in March 2020 [1]. Transmitted primarily through airborne respiratory particles [2], COVID-19 proved to be highly contagious. With an incubation period ranging from 2 to 14 days, early diagnosis of the virus presented challenges. As a result, considerable community transmission from infected individuals can occur even before the onset of symptoms. At the height of the pandemic, roughly 50% of new infections were because of transmission from asymptomatic or pre-symptomatic infected persons [3,4]. The most common viral symptoms are cough, fever, muscle ache, and shortness of breath. Mainly targeting the lower respiratory tract, COVID-19’s pathophysiology is related to acute respiratory distress syndrome (ARDS) [5]. Serious cases can lead to ARDS, systemic inflammatory reactions, or multi-organ failure [5,6,7]. Especially at risk for severe clinical outcomes are the elderly, the immunocompromised, and individuals with comorbidities or underlying health conditions [8].

Early in the pandemic, the United States Centers for Disease Control and Prevention (CDC) adopted several public health strategies designed to reduce respiratory transmission of SARS-CoV-2 [9]. Given that the primary route of the virus’ transmission is through respiratory droplets, a number of the agency’s recommendations directly targeted its airborne spread. These included social distancing guidelines, improving building ventilation systems, cautioning people to avoid nonessential indoor spaces, issuing travel guidance, and proposing universal use of face masks [9,10,11,12]. Several evidence-based studies demonstrated that wearing a well-fitting face mask or respirator lowers the risk of infection from SARS-CoV-2 [13,14,15,16,17]. Because of this perceived effectiveness, 31 states and the District of Columbia issued statewide mandates requiring face coverings by the end of July 2020 [18]. Nonetheless, masking became controversial and socially fraught.

Throughout the COVID-19 pandemic, most people in the United States recognized that wearing a face mask was an effective way to avoid SARS-CoV-2 infection [19,20,21,22,23,24]. Yet adherence to federal and state policies for wearing face masks was inconsistent. One study of over 6000 participants showed that while 90% said they wore face masks when grocery shopping, masking compliance reduced to 60% when attending religious services, declined to roughly 50% when visiting a bar or restaurant or gathering socially with others in groups of ten or more, and fell to just over 20% when visiting others’ households or exercising outdoors [22]. Other studies established that a non-insignificant minority of about 10–18% of the American populace seldom or never wore masks in public settings because of personal opposition to masking [19,21,23,25,26]. This anti-mask minority was outspoken. Nationwide, there were many anti-mask rallies and protests [27,28,29,30,31].

The reasons for non-adherence to masking policies varied. One set of reasons was health based. A sizeable number of people who refused or chose not to wear face masks said their decision came from distrust in mask effectiveness [21,25,26]. Others cited physical discomfort or negative effects such as difficulty breathing, acquiring a rash, or aggravating acne [26]. Some maintained that masks led them to overheat, suffer shortness of breath, or feel a false sense of personal health security [21]. Yet others questioned the seriousness of the COVID-19 threat or the need for masks when among people who are healthy or practicing social distancing [26,32].

Another set of reasons for noncompliance with masking policies was political–partisan affiliations, political attitudes, and politically aligned media use and behavior. Several studies highlighted the importance of partisan affiliation and political attitudes on individual behavior regarding mask wearing as well as following other health safety protocols during the COVID-19 pandemic. One study, by Calvillo, et al., concluded that political conservatives in general tended to perceive the virus as less threatening than reported in the mainstream news media, often regarding the news media’s accounts of the virus’ severity and spread as exaggerated or fake [33]. This was affirmed by opinion polling from the Pew Research Center [24]. Other studies reported that partisan affiliation, as measured by political party membership or liberal/conservative self-identification, was perhaps the single most important factor in accounting for differences in personal worry about the pandemic and decision-making about health behavior in response to it [34,35,36,37,38]. Gollwitzer, et al., found that physical distancing behavior between March–May 2020 was more strongly associated with political partisanship, as measured by county-level voting behavior in the 2016 presidential election, than a wide range of demographic characteristics—e.g., race, age, income, and residential population density [39]. Other studies showed that social distancing behavior and masking attitudes were linked, such that people espousing anti-mask attitudes often acted in disregard of social distancing guidelines [21,37]. Topazian, et al., found that attitudes toward mask wearing and social distancing were strongly correlated with individuals’ political party identity and religious practice [37]. Studies limited to mask wearing affirmed that partisan affiliation was a central factor in determining personal attitudes and behavior toward face masks [21,38,40]. One study presented evidence that, after controlling for several other variables, the practice of wearing a face mask was significantly less in counties where Donald Trump received strong support in the 2016 presidential election [41]. Empirical evidence also revealed correlations between attitudes toward mask wearing and individual consumption of news from media sources with partisan affiliation [42,43]. Specifically, many people espousing anti-mask attitudes offer support for their positions by citing news stories from conservative news media venues, such as Newsmax, Fox News, and Conservative Talk Radio [24,44]. 

Most published studies of mask-wearing attitudes and behavior were based on research conducted during the first year of the COVID-19 pandemic. One study completed in 2021 suggested that partisan differences in mask attitudes declined as the pandemic wore on [45]. However, another study from August 2023 showed that post pandemic, partisan affiliation continued to influence sentiments toward mask wearing [46]. By that time, only 12% of Americans were still wearing a mask regularly in public settings [46]. The study showed that negativity toward masking remained a partisan issue, in that among one-quarter of Republicans surveyed, 28% of individuals who describe themselves as conservatives, and 37% who identify as “very conservative” reported that they think less of people who still were wearing face masks in public [46]. This suggests that data regarding mask-wearing attitudes and behavior from the first year of the pandemic remained valid throughout its full course and beyond.

A third set of reasons for compliance or noncompliance with masking policies are normative considerations: ideological commitments, personal values, and moral virtues. This set has been the focus of a few recent research studies. The most comprehensive study, by Gelfand, et al. [38], builds off the Moral Foundations Theory (MFT) model developed by Graham, et al., in 2009 and 2011 [47,48]. According to MFT, conservatives and liberals typically emphasize different values in normative decision-making [38,47,48]. The values associated with conservatives on the MFT model include respect for authority, ingroup-loyalty, and protection of purity and sanctity [38,47,48]. Liberals, by contrast, are said to favor moral values aligned with fairness and avoiding harm/expressing care for others [38,47,48]. Gelfand, et al., used the MFT model in a moral framing experiment to try to persuade reluctant individuals to wear face masks during the COVID-19 pandemic [38]. The Gelfand study found baseline evidence showing that registered Republicans expressed less positive attitudes toward masking than registered Democrats [38]. The authors then presented seven constructed message framings, based on the MFT model, to try to promote mask wearing among both Republican and Democrat participants [38]. The results, based on multiple dependent variables, were that the framed messages had a null effect on mask-wearing behaviors [38]. The authors concluded that the “strongest effect on mask attitudes and behaviors was by far political party affiliation, such that Republicans reported having more negative attitudes and intentions toward wearing masks … compared to Democrats” [38]. The framing messages, couched in terms of the moral values identified by MFT as conservative versus liberal, had no effect in adjusting mask-wearing behavior from the stance associated with political party membership [38].

Two further studies published in 2023 similarly drew upon MFT to examine COVID-19 masking and other prophylactic behaviors. Mejova, et al. conducted a large study of Twitter messages to examine individual responses to government mask mandates in the United States during 2020 [49]. The study confirmed that anti-mask attitudes were associated most with people of conservative leanings. Yet the results did not align with what the MFT model would predict. Cautioning that their study had linguistic deficiencies, selection bias, analytic noise, and limited generalizability, the authors reported that government mask mandates resulted in a shift toward greater politicization of the masking issue. Notably, the moral values invoked by participants inexplicably diverged from those the MFT model posits as traditionally conservative or liberal [49]. The other 2023 publication, reporting on a pair of studies by Waldron et al. looked more generally at pandemic health messaging among college studies, not strictly masking behavior [50]. The authors found that moral messaging from trusted sources had a measurable effect on students’ decisions to adopt preventative health behaviors. Using exploratory factor analysis, the authors concluded that the students were influenced by messages that appealed to community responsibility, consideration for others, and self-determination. They interpreted these results as lending partial support for MFT [50]. 

Other studies have examined whether the moral virtue of empathy is associated, positively or negatively, with attitudes toward mask wearing in the context of the COVID-19 pandemic. Pfattheicher, et al., found that empathy can positively influence compliance with face mask policies as well as social distancing recommendations [51]. The authors determined that positive influence occurs from actually inducing empathy directed toward those most at risk of contracting serious illness from SARS-CoV-2 infection, while merely giving information about the importance of following COVID-19 prophylactic measures does not [51]. Another study, by Mallinas, et al., reported that empathy and the “perceived normativity” of wearing a face mask were correlated with pro-mask attitudes, independent of demographic variables including political conservatism, gender, and age [52].

The purpose of the current study was to build off and expand upon the influence of such normative considerations in individual face mask attitudes and behaviors during the COVID-19 pandemic. The aim was to determine how a number of generally overlooked normative factors and values may affect mask wearing. Specifically, the study examined the normative factors of (1) ideological consistency, (2) personal values associated with responsible news sharing and self-esteem, and (3) the moral virtue of compassion.

To study ideological consistency, the author drew upon the Pew Research Center’s Ideological Consistency Scale. The Pew Research Center developed its Ideological Consistency Scale in 1994 [53]. Since then, the Center has used it in a number of empirical research studies [54,55,56,57]. It also has been adapted for use by international researchers [58]. The Center defines Ideological Consistency as the “share of Americans who hold liberal or conservative views across a range of values dimensions; this is also sometimes referred to as ‘ideological constraint’ or ‘ideological sorting’ by political scientists and other researchers” [54]. The scale uses a set of 10 questions designed to measure the degree to which people express mostly conservative or mostly liberal points of view across a diverse range of social value dimensions. The scale does not measure the extent of ideological divides, nor does it aim to present the content of political or ideological opinions; rather, it measures coherence within an individual’s set of ideological beliefs or opinions. 

This study also sought to gain some understanding about how various aspects of individuals’ moral psychology may have influenced their mask wearing during the pandemic by measuring participants’ personal values associated with responsible news-sharing behavior and self-esteem. To measure responsible news sharing, the researcher elicited survey responses on two news-sharing parameters: the sharing of news for creating awareness and the sharing of unverified news. For self-esteem, the study used the Rosenberg Self-Esteem Scale [59,60,61]. The researcher worked from the hypothesis that an individuals’ level of responsibility in sharing unverified news, as well as their level of self-esteem may affect their compliance with face mask policies.

Finally, this study examined the moral virtue of compassion as a factor affecting personal face mask practice. Prior studies have looked at the related virtue of empathy. Compassion toward the misfortunes or conditions of distress being suffered by others is a significant moral motive for acting in socially beneficial ways. Unlike cognitive empathy, which presumes the ability to take the perspective of others, but much like affective empathy and sympathy, both of which emotively stimulate concern for the needs of others, compassion is a virtue that propels individual action in disinterested furtherance of the good of others. This study worked from the hypothesis that compassionate individuals would be more morally inclined to comply with face mask recommendations than those lacking this moral virtue. Compassion was measured using four items adapted from Gilbert, et al. [62].

In measuring these various normative considerations, the present study did not make any specific recommendations to participants regarding the use of face masks. Nor did it attempt, unlike Gelfand, et al. [38], to nudge the study participants to modify their masking behavior. Rather, the study sought to understand how ideological commitments, personality values, and the moral virtue of compassion influence individual decision-making and reported mask-wearing attitudes and behaviors among college students. Given how even now, post pandemic, little is known about what factors led to masking compliance and non-compliance, it is important to consider a wider range of variables than those commonly studied such as political party identity. Research addressing such partisan variables have provided important information about masking behavior. This study builds upon that backdrop, aiming to confirm that research while at the same time contributing to a more comprehensive understanding of the relevant variables affecting masking behavior. Specifically, this study aimed to address the following research questions: How does participant ideological consistency relate to self-reported masking behaviors? How do news-sharing behaviors relate to self-reported masking behaviors? How do compassion and self-esteem relate to self-reported masking behaviors?

## 2. Materials and Methods

### 2.1. Design

This quantitative study surveyed over 1200 college students about ideological commitments, psychological and normative values, and mask-wearing attitudes and behaviors during the COVID-19 pandemic. The study questionnaire was distributed to participants online using Qualtrics and consisted of closed-ended survey questions concerning masking attitudes and behaviors, political beliefs including ideological consistency, news-sharing behaviors, compassion, and self-esteem. The Qualtrics online survey platform, commonly used for academic research and supported by the researcher’s academic institution, was utilized because it allows for customization of question types, has flexible distribution options, and allows data to be downloaded in reports suitable for analysis. Given that data were collected during the COVID-19 pandemic and lockdown, online survey distribution was necessary. Questions pertaining to narcissism and religiosity were also included in the survey to disguise the study’s purpose but were not included in analyses. This study was approved by the researcher’s institutional review board, and the protocol met criteria for exemption with no more than minimal risks anticipated. The protocol ensured the protection of participants by requiring acceptance of the consent statement before proceeding, assuring that participation in the questionnaire was voluntary and that participants could end their involvement at any time, and providing confidentiality by removing all personal identifiers before data analysis. 

### 2.2. Participants

Participants in this study were 1231 university students recruited from Fall 2020–Spring 2021 in exchange for course credit for their participation. Before data analysis, 40 cases were excluded due to incomplete responses. Participants were aged 18 to 45 years, with an average age of 20 years (M = 19.85, SD = 1.55). The sample self-identified as 70% White, 12.3% Asian, and 5.8% Black or African American. The gender distribution was 786 (63.9%) identifying as female; 434 (35.3%) as male; 8 (0.7%) as other. Table 1 provides a summary of sample sociodemographic characteristics, and Table 2 provides information about the psychographic characteristics of respondents.

### 2.3. Measurement

Masking Behaviors. The Masking Behaviors (Self) Scale measures attitudes toward the importance of wearing a face mask to prevent the spread of illness and adherence to masking behaviors. Items include: “I believe wearing a face mask is effective in preventing the spread of COVID-19 to others”, and “I wear a face mask while in public settings”. Respondents indicated their agreement on a five-point Likert scale (ranging from ‘strongly agree’ to ‘strongly disagree’) or a frequency scale (‘always’ to ‘never’). Responses were reverse coded, resulting in higher scores on the scale showing greater acceptance and adherence to masking behaviors. The scale was reliable with a Cronbach’s alpha of 0.8427.

Ideological Consistency. The Ideological Consistency Scale is comprised of ten paired items developed by the Pew Research Center. The scale is meant to examine the degree to which individuals give mostly conservative or liberal viewpoints across a set of social/political value dimensions. One sample pair is: “Stricter environmental laws and regulations cost too many jobs and hurt the economy” (conservative position), contrasted with “Stricter environmental laws and regulations are worth the cost” (liberal position). Another pair is: “Homosexuality should be discouraged by society” (conservative position), contrasted with “Homosexuality should be accepted by society” (liberal position). Scores on this scale range from −10 (more liberal responses) to +10 (more conservative responses).

News-Sharing Behaviors. Inquiry into News-Sharing Behaviors combined two norms of personal behavior: Sharing of News for Creating Awareness, and Sharing Unverified News. Two items were averaged to measure the Sharing of News for Creating Awareness. These were: (1) “I try to create awareness by sharing news online”; and (2) “I want to educate my online friends by sharing news content online”. Cronbach’s alpha is 0.9181. Higher scores indicated greater sharing of news for creating awareness. Further, two items from Talwar, et al. [63], were used to measure Sharing Unverified News: (1) “I often share fake news because I don’t have time to check its authenticity”; and (2) “I share fake news because I don’t have time to check facts through trusted sources” [63]. These items were averaged to produce a Cronbach’s alpha of 0.9052. Higher scores indicated a greater likelihood of sharing unverified news.

Self-Esteem. Self-esteem was measured using the Rosenberg Self-Esteem Scale [59,60,61]. The scale includes ten items, such as “I take a positive attitude toward myself”, and “I am able to do things as well as most other people”. Higher scores are indicative of higher levels of self-esteem. In this study, the Cronbach’s alpha was acceptable at 0.8841.

Compassion. Compassion was measured using items adapted from Gilbert et al. [62]. The four items measuring compassion to others were: (1) “I think about and come up with helpful ways for people to cope with their distress”; (2) “I direct attention to what is likely to be helpful to others”; (3) “I take the actions and do the things that will be helpful to others”; and (4) “I express feelings of support, helpfulness, and encouragement to others” [62]. The Cronbach’s alpha for this scale was 0.7881. Higher scores on this scale indicated greater compassion to others. Means, standard deviations, and variance are presented in Table 3.

### 2.4. Statistical Analysis

SPSS Statistics version 29.0.2 was used to conduct the statistical analysis. To identify outliers, an analysis of standard residuals was conducted on the data. This indicated that six participants needed to be removed. Tests regarding the assumption of collinearity showed that multicollinearity was not an issue (Age, Tolerance = 0.99, VIF = 1.0; Social Media Hours, Tolerance = 0.72, VIF = 1.01; Internet Hours, Tolerance = 0.74, VIF = 1.35; Ideological Consistency Scale, Tolerance = 0.81, VIF = 1.23; CRS Score, Tolerance = 0.86, VIF = 1.16; Compassion to Others, Tolerance = 0.87, VIF = 1.15; Self-Esteem, Tolerance = 0.81, VIF = 1.23; Narcissism Scale, Tolerance = 0.82, VIF = 1.22; Sharing News Awareness, Tolerance = 0.69, VIF = 1.45; Corrective News Actions, Tolerance = 0.80, VIF = 1.26).

The data also met the assumption of independence of errors, such that residual terms were uncorrelated (Durbin–Watson value = 1.81). Analysis of the standardized residuals histogram showed approximately normally distributed errors, and the normal P-P plot of standardized residuals indicated that most points fell closely along a straight line. Examination of the scatterplot of standardized residuals showed that the data met the assumptions of linearity and homoscedasticity. Thus, the data fulfilled all non-zero variance assumptions (Age, Variance = 2.40; Social Media Hours, Variance = 7.32; Internet Hours, Variance = 23.89; Ideological Consistency Scale, Variance = 23.79; CRS Score, Variance = 1.18; Compassion to Others, Variance = 0.32; Self-Esteem Scale, Variance = 0.62; Narcissism Scale, Variance = 0.29; Sharing News Awareness, Variance = 1.61; Corrective Actions News, Variance = 1.01; Masking Behaviors, Variance = 0.87).

## 3. Results

Table 4 shows correlations between the main study variables of interest. There was a strong, significant correlation between the Ideological Consistency Scale and masking behaviors (*r* = −0.658, *p* < 0.001). Thus, people who exhibited greater masking behaviors tended to have more liberal social/political ideologies, whereas those with fewer masking behaviors held more conservative ideologies. There was a significant, but weak correlation between masking behaviors and sharing news awareness (*r* = 0.250, *p* < 0.001). Participants who reported sharing more news with others to create awareness had more positive attitudes toward masking. Further, there was a significant, though weak correlation between the sharing of unverified news and masking behaviors (*r* = −0.142, *p* < 0.001). Participants who reported sharing more unverified news were less likely to report mask-wearing behaviors. Though significant correlations were observed between the aforementioned variables, it is important to note that these correlations do not imply any causative relationships between variables per the survey design.

To further examine the relationship between masking behaviors and ideological consistency, participants were grouped into the following categories: Consistently Conservative (+7 to +10), Mostly Conservative (+3 to +6), Mixed (−2 to +2), Mostly Liberal (−6 to −3), and Consistently Liberal (−10 to −7). A between-subjects ANOVA was conducted with masking behaviors as the dependent variable and the Ideological Consistency Scale categories as the independent variable (see Table 5). The Sum of Squares for ideological consistency was 464.513. This indicates the amount of variation in masking behaviors due to the ideological consistency categories. The observed *F* statistic is 235.49, which is statistically significant *p* < 0.001. Thus, there is a statistically significant difference between the ideological consistency categories and masking behaviors. 

Tukey HSD multiple comparisons were used to further examine differences between the ideological consistency categories. Statistically significant differences were found between Consistently Conservative and Consistently Liberal (*p* < 0.001), between Consistently Conservative and Mixed Ideologies (*p* < 0.001), between Consistently Conservative and Mostly Liberal (*p* < 0.001), between Consistently Liberal and Mixed Ideologies (*p* < 0.001), between Mostly Conservative and Consistently Liberal (*p* < 0.001), between Consistently Liberal and Mostly Liberal (*p* < 0.001), between Mixed Ideologies and Mostly Conservative (*p* < 0.001), between Consistently Liberal and Mostly Conservative (*p* < 0.001), between Mostly Liberal and Mostly Conservative (*p* < 0.001), and between Mixed Ideologies and Mostly Liberal (*p* < 0.001). There were no significant differences between Mostly Conservative and Consistently Conservative (*p* = 0.210). Figure 1 shows estimated marginal means by ideological consistency categories.

To examine which of the aforementioned variables best predict masking behaviors, multiple regression was conducted. Masking behaviors were regressed onto ideological consistency, sharing news awareness, sharing unverified news, compassion, and self-esteem. The Adjusted Square for this model was 0.447 (Adjusted *R* Square = 0.444), which represents the amount of variance in masking behaviors that is explained by the predictors. Hence, the model explains 45% of the variance in self-reported masking behaviors, and the model is significant (*p* < 0.001). See Table 6.

In terms of variables significantly contributing to masking behaviors, ideological consistency, sharing news awareness, and sharing unverified news each made significant contributions to the model (ideological consistency *p* < 0.001, sharing news awareness *p* < 0.01, sharing unverified news *p* < 0.001). Compassion to others (*p* = 0.321) and self-esteem (*p* = 0.093) were not significant contributors, though self-esteem approached significance. 

Individually, ideological consistency contributed 33.8% of the variance in masking behaviors, sharing news awareness contributed 0.45%, sharing unverified news contributed 0.88%, compassion to others contributed 0.04%, and self-esteem contributed 0.13%. (semipartial correlation coefficients −0.581, 0.067, −0.094, 0.021, −0.036). Thus, the model significantly explains 45% of the variance in masking behaviors. Ideological consistency made the largest contribution by far, though sharing news awareness and sharing unverified news significantly contributed as well.

## 4. Discussion

Past research by other scholars found that individuals reported non-adherence to mask-wearing policies during the COVID-19 pandemic because of a range of reasons. Some people say they were reluctant to wear a face mask for health reasons related to discomfort or distrust in mask effectiveness [21,25,26]. Others attributed their masking non-adherence to their political party affiliation, personal values, or ideological self-identification [24,33,34,35,36,37,38,40]. The present study aimed to expand on this research by surveying college students to examine how ideological consistency, reported news-sharing behaviors, normative personality variables such as self-esteem, and moral virtues like compassion contribute to mask-wearing attitudes and behaviors.

Consistent with past research, participants with more liberal political ideologies reported greater adherence to masking policies while those with more conservative ideologies reported less mask-wearing behavior. This finding is not surprising given how existing research has found that masking behaviors during the COVID-19 pandemic were negatively associated with membership in the Republican party and self-identification with political conservativism [37,38,40]. Indeed, the current study indicates that ideological consistency was by far the largest predictor of an individual’s masking attitudes and behaviors among the variables examined. However, this study is unique in that it goes beyond group membership (e.g., Republican or Democrat) or simple identification with a particular labeled set of values (e.g., conservative versus liberal) as predictors of masking behavior. Rather, the study suggests that personal viewpoints across a diverse range of social and political issues (i.e., government regulation, immigration, military strength, environmental concerns, race, and sexual orientation) influence masking behaviors. Socio/political thinking is multidimensional [54]. It goes deeper and is more complex than mere partisan affiliation or political party membership. Inquiry into the consistency of ideological beliefs across a set of ten traditionally conservative versus liberal viewpoints provides richer insight into individual beliefs than people’s self-identification with a label along the political spectrum. Given that self-identification alone has been empirically found to be associated with masking attitudes and behaviors, this study’s use of the Pew Research Center’s Ideological Consistency Scale provides data that enhance our understanding of individual compliance or non-compliance with face mask policies.

The items included in the Ideological Consistency Scale go to personal beliefs concerning a set of ten paired issues of socio/political importance. Given that it was drafted thirty years ago (1994), some of the wording in the Ideological Consistency Scale seems outdated. Still, the issues the scale touches upon remain timely and are still ideologically divisive between those holding generally conservative versus traditionally liberal standpoints. Notably, the scale addresses matters of national security, immigration, individual opportunity, care for the well-being of others, personal freedom, fairness in the distribution of social goods, and the reach or overreach of governmental regulations and authority. Through this breadth of issues, the scale complements the more recent Moral Frameworks Theory (MFT) scale. MFT posits “Harm/care, Fairness/reciprocity, Ingroup/loyalty, Authority/respect, and Purity/sanctity” as the five foundational moral intuitions [48]. All of these abstract values are addressed in the more concrete social/political items on the Ideological Consistency Scale.

The Ideological Consistency Scale also dovetails with the important study of Taylor and Asmundson which used network analysis to survey adults about masking behaviors [21]. Taylor and Asmundson found that psychological reactance, combined with beliefs regarding face mask ineffectiveness, are associated with refusal to wear masks [21]. Psychological reactance is an unpleasant state that occurs when people perceive that their freedoms are threatened or restricted. Certain items pertaining to civil liberties and equality of opportunity on the Ideological Consistency Scale—e.g., items addressing job security, fairness in the distribution of social benefits, and the cost of environmental regulations—are triggers for psychological reactance.

Taylor and Asmundson reviewed a number of messaging strategies to overcome psychological reactance and adapt to mask-related reactance [21]. The strategies included stressing freedom of choice in masking messages, particularly through the use of narratives; emphasizing how individual choice affects others; forewarning potential negative experience resulting from reactance; and using reactance to strengthen pro-masking messages [21]. The present study reinforces Taylor and Asmundson’s findings. Specifically, this study suggests that emphasizing ideological positions grounded in freedom of choice may inoculate against anti-mask psychological reactance and be particularly effective when speaking to audience members with more traditionally conservative ideologies. 

This study further suggests that not only does ideological consistency predict masking attitudes and behaviors, but it provides evidence of how wide the differences can become between ideological consistencies when viewed at the categorical level. While there was no significant difference between respondents who were Mostly Conservative and Consistently Conservative in their masking behaviors, there were significant differences between the other ideological consistency categories. These differences in practices and behaviors toward wearing a face mask are shown in the estimated marginal means by ideological consistency categories depicted in Figure 1.

Although in the present study ideological consistency contributed the most to masking behaviors during the COVID-19 pandemic (33.8% of the variance), news-sharing behaviors also made a significant contribution. Sharing of news to create awareness and sharing unverified news both explain variance in overall attitudes and behaviors toward masking. However, it is noteworthy that the two news-sharing criteria corresponded to different mask-wearing attitudes and behaviors. Participants who reported sharing more news with others to create awareness evinced more positive attitudes toward masking. Those participants who reported sharing more unverified news were less likely to report pro-masking behaviors. These findings also appear to ratify past research. For instance, Hornik, et al., found that belief in misinformation about COVID-19 was negatively correlated with face mask wearing, though associations between protective behaviors and misinformation disappeared when behavioral beliefs were accounted for [64]. 

This study’s finding that unverified news sharing was related to reduced face mask-wearing behaviors highlights the importance for news media, medical organizations, and academics to continue efforts to target misinformation. The Office of the Surgeon General (OSG) identified several ways that health misinformation was addressed during the COVID-19 pandemic [65]. The OSG reported that trusted community members like physicians and educators spoke directly within their local communities about health issues. Indeed, past research supports that community members trust community leaders to provide guidance and disseminate accurate COVID-19 health information [66] and that trust in physicians and institutions positively predicts vaccine acceptance for Black and White Americans [67]. The OSG further describes that media organizations frequently identified and debunked COVID-19 misinformation. Some technological platforms monitored misinformation and attempted to limit its spread. Local, state, and the federal government sought to identify and curb major sources of misinformation, while striving to circulate accurate public health information using trusted messengers [65]. This study’s finding about a correlation between the sharing of unverified news and anti-masking behaviors reinforces the importance of the various methods of counteracting fake health-related news identified by the OSG. The relevance of this finding continues today, post pandemic, since SARS-CoV-2 remains with us as a permanent factor in the human environment.

This study further found that self-esteem and compassion for others were not significant contributors affecting compliance or non-compliance with masking behaviors during the pandemic. Though Stuppy and Smith found that people with lower levels of self-esteem showed less desire to protect themselves against contracting COVID-19 than those with higher self-esteem [68], the present study did not find that self-esteem significantly impacted masking behaviors. 

The findings of this study show that individuals’ expressed compassion for others contributed very little toward determining their mask-wearing behaviors. This finding adds importantly to previous research which suggests that ethical considerations alone did not heavily influence masking behaviors during the COVID-19 pandemic. Gelfand, et al., used moral framing messages grounded in Moral Foundations Theory (MFT) in an attempt to induce pro-mask behaviors [38]. They found that the messages had a null effect on participants’ compliance with masking policies. Mejova, et al. reported that government-imposed mask mandates had the effect of increasing the politicization of the masking issue [49]. Further, they found that the moral values which influenced the study’s participants did not align with those presumed by MFT as characteristically liberal or conservative. Waldron, et al. reported on a pair of studies examining pandemic health messaging aimed at college students [50], though their research was not specifically about masking behavior. They found that messages coming from trusted sources and appealing to self-determination, community responsibility, and consideration for others did measurably affect students’ decisions to adopt preventative health behaviors. The authors viewed these results as lending partial support for MFT [50].

While Waldron, et al. interpreted their research as providing qualified support for MFT, the Gelfand, et al. and Mejova, et al. studies found that appeals to the moral intuitions emphasized by MFT either did not affect masking behavior (Gelfand, et al.) or exacerbated the politicization of the masking issue (Mejova, et al.). Gelfand, et al., concluded that the “strongest effect on mask attitudes and behaviors was by far political party affiliation, such that Republicans reported having more negative attitudes and intentions toward wearing masks … compared to Democrats” [38]. The present study’s use of the Pew Research Center’s Ideological Consistency Scale corroborates this conclusion. Further, this study’s finding that the moral virtue of compassion has a null effect on masking behavior aligns with Gelfand, et al.’s finding that framing messages modeled on the values identified by MFT as conservative versus liberal did not shift mask-wearing behavior away from the behavior associated with political party membership [38]. Since the ten items on the Ideological Consistency Scale heavily overlap with the five more abstract values in the MFT, this study suggests that the MFT values may be less moral values than social/political values, i.e., partisan ideological predispositions. 

Finally, this study ratifies that of Pfattheicher, et al., which examined whether the moral virtue of empathy bore a relationship with individual behaviors regarding masking and compliance with social distancing recommendations [51]. Their findings were that merely informing people about COVID-19 mitigation measures did not affect masking behaviors, but that inducing empathy by stressing the risk of infecting vulnerable others with SARS-CoV-2 did impact attitudes and behaviors toward masking. Those findings—that passive information did not affect behavior while active inducement did—are confirmed by the present study. The participants here showed varying degrees of compassion for others. But those differences in their compassionate natures at best nominally affected their mask-wearing behaviors. It appears that those differences in moral nature are passive, like the mere receipt of information about COVID-19 mitigation measures reported by Pfattheicher, et al. That is, it may be that in order for the moral virtue of compassion toward others to play a significant role in influencing masking behaviors, significant efforts must be made to motivate individuals to feel compassion in an active sense toward those most at risk for serious health effects. 

## 5. Conclusions

Mask wearing is a very effective approach to reduce respiratory transmission of SARS-CoV-2. Despite its recognized effectiveness, masking became controversial during the COVID-19 pandemic and adherence to face mask policies was inconsistent. The reasons for non-adherence include factors such as health reasons, inaccuracies in risk perceptions, as well as political resistance. In particular, political conservatives may have downplayed the threat of COVID-19 severity [33,34], and there is substantial evidence that political affiliation was associated with adherence to mitigation strategies [37,39,41]. In addition, another reason for compliance or noncompliance with masking policies includes normative considerations such as ideological commitments, moral virtues, and personal values. This study examined this category of reasons. It sought to expand this category by considering ideological consistency, news-sharing behaviors, self-esteem, and compassion to determine whether these factors might overcome some of the more politically driven reasons or partisan identities that prior research has shown tend to discourage mask wearing. 

This cross-sectional online survey of college students found that socio/political ideological consistency correlates with masking behaviors, with those with weaker masking attitudes holding more conservative ideologies. Indeed, in the multiple regression model, ideological consistency was the largest predictor of masking behaviors. This finding is significant. It builds on and expands prior research by suggesting it is not simply identification with a label on the political spectrum that drives masking behaviors. Rather, this study suggests that personal viewpoints on a diverse range of social and political issues may guide masking behaviors. This finding has important public health implications, as it may be valuable for health messaging to focus on emphasizing individuals’ agency with regard to their behaviors. People who feel their freedoms are restricted may become defensive when told what to do, potentially resulting in a boomerang effect. Like past research has affirmed, it also may be important to frame messaging in terms that encourage actively helping others who are more vulnerable against potential illness. Future research should build on the relationship between ideological consistency and disease-mitigation behaviors as it appears masking motivators are not necessarily driven by political identification at a cross-section in time but rather by more long-lasting underlying views on societal issues.

While ideological consistency contributes most to masking behaviors, sharing news to create awareness and sharing unverified news also significantly contributed to variance in masking behaviors. This finding echoes the importance for academics, news media, and medical organizations to continue efforts to target misinformation because individuals who report sharing unverified news appear to have weaker mask-wearing attitudes and behaviors. Though the present study did not find self-esteem and compassion for others to significantly contribute to masking behaviors, future studies could further examine how to better motivate individuals to feel empathy or compassion toward those most at risk for health issues. 

As with other studies using college student samples, the findings in this study may not be generalizable to the entire population. It is possible that compassion and self-esteem vary across the lifespan and there could be developmental differences not adequately captured by a sample of young adults. In addition, the present sample lacks racial and ethnic diversity, so future studies would benefit from more representative samples that better capture racial and ethnic diversity. The survey design also represents self-reported data which are subject to response bias and social desirability biases. Indeed, Davies, et al., describe that self-report surveys used to examine behavior changes during the pandemic may overestimate adherence levels [69].

Although it is still uncertain which of many psychological variables may play into protective health behaviors, it is clear that political polarization seems to exert a significant role in determining masking behaviors, particularly during the COVID-19 pandemic. This study builds on the previous literature by providing evidence that politically divisive issues as addressed in the Ideological Consistency Scale also impact masking behaviors, as do news-sharing behaviors. Future research should continue to further examine these issues to improve our understanding of how to best target public health messages to diverse audience members.

## Figures and Tables

**Figure 1 ijerph-21-01652-f001:**
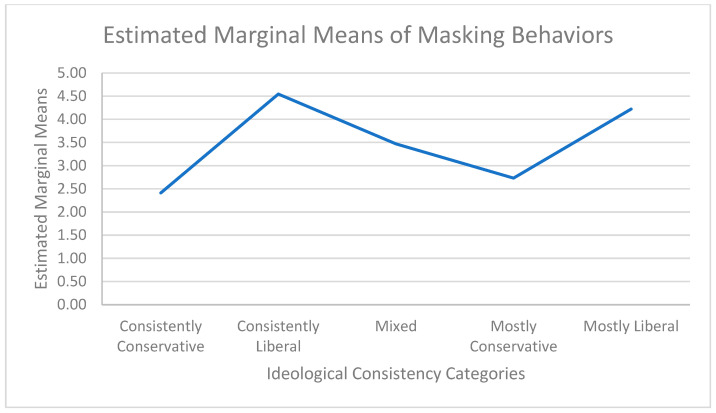
Estimated marginal means of masking by ideological consistency.

**Table 1 ijerph-21-01652-t001:** Sociodemographic characteristics of participants (*n* = 1231).

Participant Characteristics	*n*	%
Gender		
Female	786	63.9
Male	434	35.3
Other	8	0.7
Race and Ethnicity		
American Indian or Alaskan Native	1	0.1
Asian	152	12.3
Black of African American	72	5.9
Hispanic or Latino/a	63	5.1
Native Hawai’ian or Other Pacific Islander	5	0.4
Non-Resident Alien	4	0.3
Two or more races	68	5.5
White	862	70.0
Ethnicity Unknown	3	0.2
Year in College		
Freshman	233	18.9
Sophomore	539	43.8
Junior	293	23.8
Senior	164	13.3
Graduate Student	2	0.2
Age		
18	114	9.3
19	426	34.8
20	406	33.1
21	188	15.3
22–45	92	7.5
Religion		
Agnostic	107	8.7
Atheist	81	6.6
Buddhist	13	1.1
Hindu	33	2.7
Jewish	25	2.0
Muslim	23	1.9
Protestant	253	20.6
Orthodox	13	1.1
Roman Catholic	271	22.0
Something Else	145	11.8
Nothing in Particular	265	21.5

Percentages may not add up due to rounding.

**Table 2 ijerph-21-01652-t002:** Psychographic characteristics of participants (*n* = 1231).

Participant Characteristics	*n*	%
Daily Internet Hours		
0–2.0	85	7.0
2.01–4.0	210	17.2
4.01–6.0	331	27.1
6.01–8.0	269	22.0
8.01–10.0	188	15.4
10.01+	137	11.2
Daily Social Media Hours		
0.0–1.0	219	17.9
1.01–2.0	303	24.8
2.01–3.0	282	23.1
3.01–4.0	168	13.7
4.01–5.0	124	10.1
5.01+	127	10.4
Political Party Identification		
Weak Republican	76	6.2
Lean Republican	194	15.8
Strong Republican	97	7.9
Independent	285	23.2
Weak Democrat	167	13.6
Lean Democrat	218	17.7
Strong Democrat	193	15.7
Political Ideology		
Extremely Conservative	11	0.9
Very Conservative	103	8.4
Somewhat Conservative	219	17.8
Moderate	325	26.4
Somewhat Liberal	263	21.4
Very Liberal	239	19.4
Extremely Liberal	71	5.8

Percentages may not add up due to rounding.

**Table 3 ijerph-21-01652-t003:** Descriptive statistics.

Variables	Mean	Standard Deviation	Variance
Ideological Consistency Scale	−4.43	4.88	23.79
Compassion to Others	4.21	0.56	0.32
Self-Esteem Scale	3.71	0.79	0.62
Sharing News Awareness	2.90	1.27	1.61
Sharing Unverified News	1.60	0.86	0.74
Masking Behaviors	4.02	0.93	0.87

**Table 4 ijerph-21-01652-t004:** Correlation table.

	Age	SM	I	IC	CR	C	S-E	N	SN	SU	MB
Age	1										
SM	−0.031	1									
I	0.050	0.491 **	1								
IC	−0.002	−0.008	−0.065 *	1							
CR	−0.025	0.061 *	−0.033	0.269 **	1						
C	−0.013	0.007	0.001	−0.050	0.212 **	1					
S-E	0.056 *	−0.033	−0.036	0.152 **	0.130 **	0.173 **	1				
N	−0.025	0.064 *	0.074 **	−0.128 **	−0.060 *	−0.187 **	−0.387 **	1			
SN	−0.052	0.195 **	0.047	−0.297 **	0.013	0.150 **	−0.125 **	0.081 **	1		
SU	−0.010	0.108 **	−0.018	0.090 **	0.022	−0.156 **	−0.079 **	0.097 **	0.153 **	1	
MB	0.064 *	−0.015	0.094 **	−0.658 **	−0.160 **	0.071 *	−0.129 **	0.078 **	0.250 **	−0.142 **	1

ns = not significant (*p* > 0.05), * *p* < 0.05, ** *p* < 0.01; Age = Age (in years); SM = Social Media Hours (Daily); I = Internet Hours (Daily); IC = Ideological Consistency Scale; CR = CRS Score; C = Compassion to Others; S-E = Self-Esteem Scale; N = Narcissism Scale; SN = Sharing News Awareness; SU = Sharing Unverified News; MB = Masking Behaviors.

**Table 5 ijerph-21-01652-t005:** ANOVA table.

	SS	df	*F*	*p*	Partial *η*^2^
Ideological Consistency Categories	464.51	4	235.49	*p* < 0.001	0.434

**Table 6 ijerph-21-01652-t006:** Multiple regression analysis predicting masking behaviors with ideological consistency, sharing news awareness, sharing unverified news, compassion, and self-esteem.

Predictor	B	SE B	Beta	sr	*p*
Ideological Consistency	−0.119	0.004	−0.621 **	−0.581	*p* < 0.001
Sharing News Awareness	0.053	0.017	0.073 *	0.067	*p* < 0.01
Sharing Unverified News	−0.106	0.024	−0.097 **	−0.094	*p* < 0.001
Compassion	0.037	0.037	0.022	0.021	*p* = 0.321
Self-Esteem	−0.044	0.026	0.022	−0.036	*p* = 0.093

sr = semipartial correlation coefficient. * *p* < 0.01, ** *p* < 0.001.

## Data Availability

The original contributions presented in the study are included in the article. Further inquiries can be directed to the corresponding author.

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
