# Peer review of "Ideological Consistency and News Sharing as Predictors of Masking Among College Students"

_ijerph, 2024, doi:10.3390/ijerph21121652_

Round 1
Reviewer 1 Report
Comments and Suggestions for Authors
Overall: This is a well conducted and described study, which is largely ready for publication. I just have a few minor tweaks for consideration below:
- 6: First line of abstract has extra COVID-19
- 13: sure, ideological consistency, but perhaps we want to know in the abstract the direction of that ideological lean?
- 26: Jumping tense here
- 42: I’m not sure about universal here. While I don’t have the original CDC guidelines in front of me, I would take ‘universal’ to mean ‘all people, all the time’, which it wasn’t.
- 42: Is the ‘contentious’ sentence in the right place?
- 61-71-110 I wonder if you could disentangle / state which of these are vocalised (perhaps disingenuous) reasons vs underlying reasons.
- 142-144: tense shift
- 404: some of the percentages in this paragraph seem wrong - eg it should probably be 45% (or .45), not .45%
- 482: I might be confused by your data, I didn’t think your results actually showed anything associating mask policies (rather than behaviours) with fake news sharing
- 487: Sure, people did this, but was it effective?
• ⁃ 499: There is no such thing as “approaching significance”, I don’t see how this modestly supports Stuppy and Smith’s work
Reviewer 2 Report
Comments and Suggestions for Authors
The research goal is clearly stated, and the literature review is well presented. Regarding the research methodology, the design of the study is well presented. The authors should formulate more clearly their research hyprotheses. When presenting results it should be plainly explained that correlation doesn’t mean that there is a causal relationship between variables.
Reviewer 3 Report
Comments and Suggestions for Authors
Although the study addresses a topic that emerged nearly four years ago, it successfully leverages empirical data to explore public health issues and their socio-political and cultural dimensions. The study employs a coherent methodology and is presented in a well-organized and persuasive manner. However, there are several observations that could help the author further enhance the quality and impact of the paper.
Title
1- Given that the study’s sample is limited to college students, it would be better to revise the title to reflect the specific population studied.
Abstract
2- The first sentence of the abstract would get further revision to enhance clarity. Currently, it provides a general context for the study but could be streamlined to immediately highlight the specific focus and significance of the research.
3- It would be preferable to avoid using the acronym "CDC" in the abstract, as it may not be immediately recognizable to all readers. Instead, the full name, "Centers for Disease Control and Prevention," should be used to ensure clarity and accessibility, particularly for an international audience.
4- The abstract should highlight the significance of the study, particularly given its timing long after the height of the COVID-19 pandemic. It would be valuable to show how the findings contribute to current public health strategies or understanding of socio-political dynamics.
Introduction
5- The connection between the 2016 US presidential election and COVID-19 physical distancing behavior requires further clarification in the manuscript. Specifically, the relationship described in the sentence, "Gollwitzer, et al., found that physical distancing behavior was more strongly associated with political affiliation, as measured by voting behavior in the 2016 presidential election, than a wide range of demographic characteristics – e.g., race, age, income, and residential population density," is unclear. The author needs to elaborate on how voting behavior from the 2016 election serves as a relevant proxy for political affiliation during the pandemic in 2020.
6- The statement “A third set of reasons for compliance or noncompliance with masking policies are normative considerations: ideological commitments, personal values, and moral virtues. This set has been understudied” could be reconsidered, as there is already a body of literature that examines such topics. For instance, the following studies provide valuable insights into the normative and moral dimensions of masking behaviors:
- Mejova, Y., Kalimeri, K., & De Francisci Morales, G. (2023). Authority without Care: Moral Values behind the Mask Mandate Response. Proceedings of the International AAAI Conference on Web and Social Media, 17(1), 614–625.
- Abbott, O., May, V., Woodward, S., Meckin, R., & Gilman, L. (2023). Masks, Lay Moralities, and Moral Practice. In Masking in the Pandemic: Materiality, Interaction, and Moral Practice (pp. 61–86). Cham: Springer Nature Switzerland.
- Ji, P. (2020). Masking morality in the making: how China’s anti-epidemic promotional videos present facemask as a techno-moral mediator. Social Semiotics, 33(1), 232–239. https://doi.org/10.1080/10350330.2020.1810462
- Waldron, V. R., Reutlinger, C., Martin, J., O’Neil, E., & Niess, L. C. (2023). “We Are All in This Together”: Which Memorable Moral Messages Guided Student Responses to the COVID-19 Pandemic? Health Communication, 39(12), 2744–2755. https://doi.org/10.1080/10410236.2023.2286695
7- Among the numerous studies on this topic, the author should include a clear paragraph detailing the contributions of this study to the existing literature. Rather than stating that the topic is "under-studied," it would be more effective to highlight the specific gaps this research addresses and its relevance, especially given its publication nearly four years after the pandemic's peak.
8- The author needs to provide a clear and precise definition of ideological consistency, outlining what it specifically means in the context of this study. Additionally, it would be better to explain how this concept is operationalized and integrated into the research framework
9- The paper would benefit from updating, particularly given that the data was collected three years ago. The author should consider how this dataset can be reanalyzed or contextualized to offer fresh insights or provide new knowledge that remains relevant in the current public health and socio-political environment.
Materials and Methods
10- The study's reliance on a survey administered via the Qualtrics platform necessitates a more detailed explanation of the platform's selection.
11- The study's methodology section requires further elaboration on ethical considerations and data privacy measures, particularly concerning the collection of sensitive personal information such as race, religion, and other identifiers. While the paper mentions obtaining approval from the researcher's institutional review board, it lacks specifics about the approval pertains exclusively to this study or a broader project. Clarification is needed on whether the institutional review board approval explicitly covered the collection of personal data from individuals.
12- Regarding measurement, the study identifies “sharing of fake news” as a factor influencing masking behaviors and measures it using items that assess participants’ acknowledgment of sharing news without verifying its authenticity. However, there appears to be a conceptual distinction between “sharing fake news” and “sharing news without verification.” The former implies the intentional or unintentional dissemination of false or misleading information, while the latter focuses on a lack of verification effort, which may or may not result in sharing false information. This distinction is important because “sharing fake news” involves an outcome (falsehood of the news), whereas “sharing without verification” reflects a process or behavior. As a result, the current measurement might capture a broader construct that includes sharing true but unchecked news, potentially diluting the specific impact of fake news dissemination on masking behaviors. To strengthen the validity of this factor, the measurement items could be refined to explicitly capture the sharing of demonstrably false or misleading information. Alternatively, the construct could be renamed to “sharing unverified news” to better align with the current measurement approach.
Round 2
Reviewer 3 Report
Comments and Suggestions for Authors
It is commendable that the author has thoroughly incorporated the suggestions, resulting in an article that is now more structured and readable.